# Chemical Effect of Bisphenol A on Non-Alcoholic Fatty Liver Disease

**DOI:** 10.3390/ijerph16173134

**Published:** 2019-08-28

**Authors:** Marcello Dallio, Nadia Diano, Mario Masarone, Antonietta Gerarda Gravina, Vittorio Patanè, Mario Romeo, Rosa Di Sarno, Sonia Errico, Carla Nicolucci, Ludovico Abenavoli, Emidio Scarpellini, Luigi Boccuto, Marcello Persico, Carmelina Loguercio, Alessandro Federico

**Affiliations:** 1Department of Precision Medicine, University of Campania “Luigi Vanvitelli”, via Pansini 5, 80131 Naples, Italy; 2Department of Experimental Medicine, University of Campania “Luigi Vanvitelli”, via Pansini 5, 80131 Naples, Italy; 3Department of Medicine and Surgery, University of Salerno, via Salvador Allende, 84081 Salerno, Italy; 4Department of Health Sciences, University “Magna Graecia”, Viale Europa–Germaneto, 88110 Catanzaro, Italy; 5Division of Gastroenterology, Department of Internal Medicine, TARGID, University Hospital Gasthuisberg, 3000 Leuven, Belgium; 6Greenwood Genetic Center, 113 Gregor Mendel Circle, Greenwood, SC 29646, USA

**Keywords:** bisphenol A, endocrine-disrupting compounds, oxidative stress, non-alcoholic fatty liver disease, hepatocellular carcinoma

## Abstract

Non-alcoholic fatty liver disease (NAFLD) is considered a predominant chronic liver disease worldwide and a component of metabolic syndrome. Due to its relationship with multiple organs, it is extremely complex to precisely define its pathogenesis as well as to set appropriate therapeutic and preventive strategies. Endocrine disruptors (EDCs) in general, and bisphenol A (BPA) in particular, are a heterogeneous group of substances, largely distributed in daily use items, able to interfere with the normal signaling of several hormones that seem to be related to type 2 diabetes mellitus (T2DM), obesity, and other metabolic disorders. It is reasonable to hypothesize a BPA involvement in the pathogenesis and evolution of NAFLD. However, its mechanisms of action as well as its burden in the vicious circle that connects obesity, T2DM, metabolic syndrome, and NAFLD still remain to be completely defined. In this review we analyzed the scientific evidence on this promising research area, in order to provide an overview of the harmful effects linked to the exposure to EDCs as well as to frame the role that BPA would have in all phases of NAFLD evolution.

## 1. Introduction

A large amount of studies produced in the scientific literature on this topic in recent decades, emphasized the epidemiological importance of non-alcoholic fatty liver disease (NAFLD) [1]. NAFLD will be the most important cause of chronic liver disease in adults and children and it could become the leading indication for liver transplantation [1]. In the last 30 years, its prevalence has markedly increased, generating the necessity by the scientific community to produce studies focusing the attention on new risk factors promoting its onset [2]. NAFLD is a spectrum of diseases that ranges from simple steatosis to steatohepatitis (NASH) with a potential risk of progression toward cirrhosis and hepatocellular carcinoma (HCC) [3]. It is part of the metabolic syndrome and consequently it is associated with obesity, insulin resistance (IR), and type 2 diabetes mellitus (T2DM) [4].

In the picture of NAFLD, the diagnosis of NASH involves principally the 40 to 50 years age group, whereas for the diagnosis of NASH-related cirrhosis the age group more involved is 50 to 60 years. However, the emerging obesity epidemic resulted in an increased number of children with this disease, sometimes with advanced fibrosis [5]. NAFLD originates from the interaction between environmental exposure and a susceptible polygenic background (about 20% of patients report a family history of unexplained liver disease) and comprises multiple independent modifiers [6]. Despite the fundamental mechanisms governing the disease progression are shared among all NAFLD patients, there is an inter-individual variability. This may be due to the patient’s genetic background, the gut microbiome composition, and/or pollution exposition, which all of them can modulate different clinical phenotypes [6].

Due to the development of industrialized areas and the request of fast and cheap foods, the exposition to chemical substances able to trigger or worsen several diseases is increasing throughout the world. 

This phenomenon is growing in Western countries, opening the way to the possibility to observe an important change in both the epidemiology and the natural history of chronic liver diseases [7,8,9]. 

Endocrine-disrupting compounds (EDCs) (Table 1) are a heterogeneous group of substances able to interfere with physiological hormonal signaling thus increasing the risk of developing several pathologies, in particular metabolic ones [10]. 

Over the years, the amount of scientific evidence proving a direct correlation between EDCs exposure and adverse health effects, including neurodegenerative, cardiovascular, bone and immune diseases, and disorders of the reproductive or metabolic system, has exponentially grown [11,12]. 

Their role in several pathological conditions is now widely accepted, however the complete definition of their mechanisms of action is still unsolved, thus representing an arduous effort for scientists to unveil the underlying molecular processes leading to disease-onset. 

However, in this wide scenario, it is reasonable to suppose that their fundamental activity is to mimic or interfere with endogenous hormonal stimulation in different steps of physiological molecular signaling (biosynthesis, release, receptor interaction, downstream signaling, and degradation) of thyroid, hypothalamic, and sexual hormones principally [13]. It was thought that EDCs exert their actions by interacting with nuclear hormones receptors as well as with several types of cellular receptor such as non-nuclear steroid hormone, non-steroid hormone, and orphan receptors. They could be able to interfere with biological homeostasis not only by acting on steroid biosynthesis and metabolism but also by regulating neurotransmission of central and peripheral nervous systems [13,14]. 

In the last few decades, the list of the substances classified as EDCs has dramatically increased. Moreover, their spreading as constitutive elements of common products has become a global research priority in particular in the chronic metabolic diseases setting [11]. Considering their chemical origin, EDCs are both natural (e.g., isoflavone a phytoestrogen) and synthetic compounds like industrial solvents, dioxins, solid metalloids (e.g., arsenic) or particulate matter (PM), but also agricultural pesticides e.g., dichlorodiphenyltrichloroethane (DDT), tolylfluanid (TF—a pesticide used on fruit and ornamental plants), bisphenol A (BPA—a food contaminant widely used to manufacture polycarbonate plastics and epoxyresins), pharmaceutical agents and many others recurrent products. Furthermore, a large part of these compounds has been recently renamed “persistent organic pollutants” (POPs) to underline their ability of being resistant to environmental degradation and, thereby, depositing in human and animal tissues, in the case of continuous exposure [13].

Given that, the exposition to the aforementioned substances concerns not only certain groups of people (e.g., worker vs. non-worker) or circumstances (rural, city, industrialized areas) but also the greater part of the population, due to the ubiquitary diffusion of several daily use products containing high concentration of EDCs, with potentially severe harmful effects on human health [15,16]. In this review we elaborated an overview on the molecular mechanisms of EDCs, in particular BPA, surrounding the pathogenesis of non-alcoholic fatty liver disease.

### The Exposition and Pharmacokinetics of BPA

Focusing the attention on BPA we must highlight the concept of over-exposition to this EDC due to its large diffusion. To give an insight on the importance of this concern, every year United States (US) chemical industries produce more than 6 million pounds of BPA, whereas the US environmental protection agency (EPA) has classified it as the third most important environment contaminant [17]. 

It represents not only an important public health problem but also one of the most important factors able to increase the national health expenditures, due to the increase of new cases of BPA induced obesity/T2DM with an estimated healthcare cost of about 1.54 billion euros [18]. 

BPA (4,4-isopropylidenediphenol) is an organic compound characterized by two phenol rings connected by a methyl bridge, with two methyl functional groups, classified in the group of phenols. Due to its high resistance to a wide range of temperatures and acids, as well its hardness and transparency, this compound is extensively used in industry as a material for the production of phenol resins, polyacrylates and polyesters, and principally for the production of epoxy resins and polycarbonate plastics that are used for the manufacture of everyday products [19]. 

The most relevant routes of exposure to BPA are represented by environmental pollution, ingestion, inhalation, or dermal contact. Anyway, among them, the most important is represented by eating and/or drinking contaminated food, canned dietary products, pre-packed foodstuffs, packaged baby formula, baby bottles, and containers used for food-storage [17]. 

The use of all the above mentioned items determines an average exposure to BPA that has been estimated to be in a range from < 1 to 5 μg/kg of body weight (BW)/day [19]. Unfortunately, it is difficult to establish the minimum toxic dose capable of exerting harmful effects [19]. 

Moreover, there seems to be a difference regarding minimum toxic dose in animals and in humans. In animal models, the endocrine disruptive effects of BPA become clinically relevant at a dosage of 1 μg/kg BW/day, even if some scientific evidence demonstrated that lower expositions could be sufficient to lead to several metabolic consequences. Otherwise, in human clinical models, the daily safe oral dose of this compound has been set at 5 μg/kg body weight/day [20,21,22]. After oral absorption, BPA is glucuronidated by the liver, distributed in a lot of human tissues and eliminated mainly through kidney excretion [19]. 

Despite this simple and common metabolic pathway, BPA has been also detected in amniotic fluid, follicular fluid, placental tissue, umbilical cord blood, and also in breast milk [23]. In particular, the presence of this EDC in maternal and fetal serum, as well as in breast milk, may suggest a possible role of long-term exposure to BPA during the fetal and neonatal period on long-term harmful effects on the new-born [19]. In addition, in this setting, a recent study suggests a possible phenomenon of bioaccumulation in the adipose tissue, due to the great affinity of BPA for fatty acids. According to this possibility, a previous study of our group demonstrated, in a group of NAFLD patients after a one-month BPA-free diet, a reduction of circulating plasmatic BPA levels in absence of a significant reduction of urinary levels in comparison to baseline [24]. This phenomenon could be produced by the continuous release of BPA from the adipose tissue reservoir, which in turn could explain the reduction of BPA in plasma but not in urine after a period of BPA-free diet. This could be due to the fact that BPA may be able to saturate the kidney excretion already at low concentrations, leading to its decrease only in plasma (and therefore in a clinically irrelevant manner) after only one month of BPA free diet [24].

This finding could be clinically relevant for the possible onset of BPA related disorders even after brief exposure suspension, nevertheless, more studies are necessary to quantitatively evaluate the “BPA adipose storage”.

## 2. EDCs and Metabolic Diseases 

The aforementioned considerations lead to underlining the existing link between EDCs, their widespread diffusion, and their potential human health harmful effects. Several epidemiological and experimental studies suggested a strong relationship between EDCs exposure and the pathogenesis of Insulin Resistance (IR) associated diseases (a part from T2DM), such as poly-cystic ovary syndrome (PCOs), non-alcoholic fatty liver disease (NAFLD), and obesity [10,11,12,13].

In the last decades the incidence and prevalence of metabolic diseases has spread worldwide. EDCs exposure appears to be more dangerous in specific lifetime periods. In particular, an early lifespan exposure is supposed to be more dangerous, not only by leading to an immediate onset of specific pathologies, but also because it can make the exposed subject more prone to develop several metabolic diseases in the future, even without further significant exposure to EDCs, due to heritable and epigenetic changes [10,25]. 

In particular, a large part of EDCs seem to be able to induce alterations in lipid and carbohydrate metabolism [26]. The arsenic (a solid metalloid included in EDCs) exposure appears to be related to lower levels of high-density lipoprotein (HDL) and higher levels of oxidized low-density lipoproteins (O-LDL), despite lower total levels of LDL cholesterol (LDLc) and total cholesterol [10]. 

Considering the EDCs main capability to interfere with endocrinal pathways, it is not surprising that the greater part of them could be able to interfere with glucose homeostasis promoting the onset of IR and T2DM. Various modes of interference have been proposed, involving, among others, the disruption of the pancreatic beta-cells insulin production and the peripheral insulin activity, as well as the reduction of insulin receptor substrate-1 (IRS-1) expression and Protein kinase B-2 (Akt2) phosphorylation [10]. 

In this regard, PM can interfere with insulin glucose fasting homeostasis in vitro (specifically in a reverse relationship with its size) as well as the TF is able to impair insulin signaling in primary rodent and human adipocytes through a reduction in IRS-1 levels [10,27]. 

All these topics are relevant in the setting of metabolic health because they surround the context of metabolic syndrome that in turn represents the most important feature to explain the pathogenesis of NAFLD, as well as they represent the most important factors able to cause the worsening of this disease in more advanced stages.

### Mechanisms of Biological Action of BPA 

As already mentioned above, chronic exposure to low levels of BPA significantly affect biological systems through the disruption of hormonal homeostasis and cell signaling including, above all, inflammation or immune response-related, reproductive-related, cancer-related, and nervous system cell signaling pathways [19] (Figure 1). 

Since several studies have suggested that high (serum or urinary) levels of BPA could be associated with obesity, IR, and metabolic syndrome and all of these conditions have also been related to systemic inflammatory responses, it seems currently fascinating to focus on the emerging role of BPA exposure in the metabolic and endocrine cell signaling pathways’ alterations in order to explain the pathogenetic link between environmental pollution and the development of chronic metabolic liver diseases [28,29]. BPA probably acts using an indirect way, linking nuclear and membrane receptors such as estrogen receptor α/β (ERα/β), androgen receptor (AR), G protein-coupled estrogen receptor (GPER)—also known as G protein-coupled receptor 30 (GPR30)—, insulin-like growth factor-1 receptor (IGF-1R), and estrogen-related receptor gamma (ERRγ), reducing the biological response related to their activation by their physiological ligands. However, this compound could also exploit its effects directly through a BPA-mediated stimulation of these receptors [30,31,32,33,34].

Concerning GPR30, its activation induces ERK/mitogen-activated protein kinase (MAPK) and phosphatidylinositol-3 kinase (PI3K)/AKT pathways through a “non-genomic mechanism” [12]. This receptor has become notorious for transducing the estrogen-induced signal through the classic mechanism of G protein-coupled receptors. Moreover GPR-30, by means of the subunits β/γ is able to transactivate the epidermal growth factor receptor (EGFR); this receptor can thus activate the MAPK pathway or the PI3K/AKT ones. This interaction can cause the activation of MAPK and PI3K pathways in the immune cells, enhancing the proinflammatory mediators’ production, such as tumor necrosis factor alpha (TNF-α), interleukin (IL)-1 and IL-6 that are some of the most important inflammatory mediators that sustain the process of systemic inflammation linked to the metabolic syndrome [19]. As known, the systemic inflammation determines a clear worsening of the IR and therefore enhances the pathogenetic link with metabolic diseases, generating a vicious cycle able to worsen the metabolic homeostasis (Figure 2). 

BPA activates GPR30. In the immune cells, GPR 30 mediated activation of MAPKs and PI3Ks pathways (through EGFR transactivation) induces the production of inflammatory cytokines; in turn, the inflammation worsen the IR, leading to an increase in the demand for insulin by tissues; this determines a consequent increase in inflammation with negative implications on peripheral insulin sensitivity, thus triggering a vicious circle.

BPA, bisphenol A; GPR30, G protein-coupled receptor 30; EGFR, Epidermal Growth Factor Receptor; ERK, extracellular signal-regulated kinase; RAS, Rat sarcoma; RAF, proto-oncogene serine/threonine-protein kinase; PI3K, phosphatidylinositol-3 kinase; Akt, kinase inhibiting apoptosis; mTOR, mammalian target of rapamycin; TNF-α, Tumor Necrosis Factor; IL-1, Interleukin 1; IL-6 Interleukin-6; IR, insulin resistance.

Soriano et al. demonstrated that mice exposed to 1 nM of BPA developed diabetes with a peculiar mechanism involving BPA mediated activation of ERα and altered function of K-ATP channels leading to the dysregulation of insulin release by pancreatic β cells [35]. In addition, Kang et al. used a mouse model of diabetes involving streptozocin-induced insulin-deficiency, in which BPA exposure, through the expression of pancreatic ERα and inflammatory-related cytokines (including TNF-α and IL-1), increased serum insulin levels determining a worsening of IR [36]. 

However, more recently, BPA has been proven to be capable to act as an E2 (17β-estradiol) mimetic compound in presence of ERα, by linking the ERα receptor, leading to an increase in the expression of glucose transporters (GLUT)-4 and also in the glucose uptake [12]. 

Even though this last effect seems to be paradoxical and in contradiction with an IR condition, it could be justified by the fact that the increased intracellular glucose levels can consequently induce fast cellular proliferation (with an interesting view to a neoplastic context). Moreover, it seems to be obligatory to focus the attention on the possibility that BPA is able to interact with several types of receptor families and, among them, with a lot of subtypes, determining different biological effects. The aforementioned evidences suggest a BPA-predominant role in inflammation processes. Nevertheless, especially in the description of the metabolic liver diseases, inflammation does not have to be considered as the only one involved: the closely related oxidative stress as well as mitochondrial dysfunction have to be considered too. 

In this regard, a part of BPA-induced oxidative stress mechanisms have been recently clarified: while a majority of BPA is converted into less toxic metabolites, the remaining free BPA induces reactive oxygen species (ROS) production through the enzymatic (H2O2/peroxidase and NADPH/CYP450) and non-enzymatic (ONOO−/CO2 and −OCl/HOCl) reactions [37]. 

These mechanisms contribute to create an oxidative microenvironment where liver cells can be directly injured or indirectly damaged through ROS-induced cellular defenses down regulation [37] (Figure 3). 

BPA is converted (for the most part) into less toxic metabolites; the remaining free BPA induces the enzymatic (H2O2/peroxidase and NADPH/CYP450) and non-enzymatic (ONOO−/CO2 and −OCl/HOCl) formation of phenoxyl radicals that interact with NADPH or intracellular GSH along with further enzymatic processing, producing thus a variety of radical species, including superoxides, peroxides, and hydroxyl radicals. In this oxidative microenviroment, liver cell injuring and down regulation of the cellular defenses are possible. BPA, bisphenol A; CYP450, cytochrome P450; GSH, Glutathione; H2O2, hydrogen peroxide; ONOO, peroxynitrite; CO2, carbon dioxide; OCl- hypochlorite ion; HOCl, hypochlorous acid.

BPA exposure seems be able to decrease the anti-oxidative enzyme catalase (CAT), by interacting with it through an hydrophobic and electrostatic effect, inducing a structural change and a consequent loss of function of the enzyme [38].

In this regard, a recent study on male wistar rats demonstrated how a chronic-low-dose BPA oral administration could disrupt oxidative stress homeostasis [39]. In this animal model, BPA seems to increase liver cell malondialdehyde (MDA) levels, a lipid peroxidation marker, and contemporarily reduce glutathione (GSH), the main antioxidant defense of the hepatocytes [39]. Moreover, in the same animal model, a 30-day BPA administration at a dosage of 25 mg/kg, decreased also CAT and superoxide dismutase (SOD) activity levels [40]. Interestingly, the authors also evaluated liver levels of miRNAs (MiR) regulating the production of several inteleukins: pro-inflammatory (TNF-α, IL-1, IL-6) and anti-inflammatory ones (IL-10) demonstrating an increase of the first group and a decrease of the second. This finding highlighted, one more time, the direct link between oxidative stress induction and the generation of an inflammatory microenvironment that represents the pathogenetic key to explain the worsening of NAFLD in this setting [40]. 

Therefore, the metabolic disorders associated to BPA exposure (obesity, T2DM, PCOs, metabolic liver diseases) seem to be characterized by a common pathogenetic denominator identified in BPA-induced inflammation with IR and oxidative stress. However, in addition to these elucidated mechanisms, a more complex mode of action of BPA, involving gut microbiome as well as various neurotransmitter systems, is emerging. In this regard, BPA-perinatal exposed mice demonstrated a high risk of developing obesity and metabolic disorders in adult life due to dysregulation of inflammatory response induced by a modification in bacterial intestinal flora, which consisted of the decrement in the proportion of *bifidobacterium* and *firmicutes* population [18].

Regarding the interference of EDCs on the nervous system, a recent study has shown the BPA capability to interfere with the parasympathetic innervation of the liver in a porcine model [17]. This neurochemical disruption modifies the levels of neuropeptides, such as G-alanine and g-alanine-like peptide (GAL), cocaine- and amphetamine-regulated transcript (CART), and Calcitonin Gene Related Peptide (CGRP), that control the long-term hunger and food intake signaling mechanism conducing thus to childhood obesity and diabetes development [17]. Concerning the endocannabinoid system, adult zebrafish exposed to BPA showed an increase in the liver levels of the obesogenic endocannabinoids 2-arachidonoylglycerol (2AG) and anandamide (AEA) and a concomitant decrease in palmitoylethanolamide (PEA) [41]. Moreover, the main AEA hydrolytic enzyme activity decreased (i.e., fatty acid amide hydrolase), while the expression of the endocannabinoid receptor 1 (CB1) increased. All these changes determined liver steatosis inducing triglyceride (TG) accumulation in a CB1-dependent manner [41].

## 3. NAFLD and EDCs 

NAFLD is strongly linked to over nutrition, under activity, and IR, but many other factors initiating hepatic steatosis or supporting the progression to NASH have been proposed [42]. These include biologic or synthetic hepatotoxins, bacterial endotoxins, and exposure to EDCs [43]. A combined biochemical (hepatic lipotoxicity) and immunological (adipokine-mediated) mechanism acting contemporarily could drive NAFLD liver injury. Hepatic lipotoxicity develops as an adaptive mechanism that leads to ROS generation, ER stress, and cellular dysfunction [6]. Finally, cellular damage triggers a mixture of immune-mediated hepatocellular injury with the activation of both necrotic and apoptotic cell death pathways. Once these mechanisms persist, they cause stellate cell activation, fibrogenesis, and disease progression [6]. However, inappropriate regulation of hepatic de novo lipogenesis is now believed to facilitate the generation of lipotoxic lipid intermediates that could contribute to the pathogenesis of NASH [43]. In NAFLD, the prevalence of HCC is estimated to be 0.5%, but it increases to 2.8% in patients with NASH. Nowadays liver biopsy, furthermore, is the only procedure that reliably distinguishes NAFL from NASH, but there are practical limitations, including costs and risks [6]. The biggest advantage in practicing liver biopsy is the determination of fibrosis level, the key histological determinant of overall and liver specific mortality in NASH [44]. High-quality prospective data on the progression of NAFLD are limited, especially in primary care. A meta-analysis demonstrated that NAFLD increases all-cause mortality [45]. Patients with steatosis with no additional features of liver injury may follow a generally benign course, with mortality similar to that of age- and sex-matched controls [46]. In another cohort study, NASH was associated with > 10-fold increased risk of liver-related death (2.8% versus 0.2%) and death from cardiovascular disease over a mean follow-up period of 13.7 years [47]. Given the previously cited evidence, it has been largely shown how BPA is related to several metabolic disorders and it seems to be involved in every step of NAFLD onset and disease progression too.

### 3.1. Hepatic TG Accumulation 

In the liver, BPA exposure seems to be able to determine TG accumulation through a multifactorial way including IR, increased de novo lipogenesis (DNL), and lipidic homeostasis disruption [43,48].

As discussed above, BPA exposure determines IR through various mechanisms. IR, centrally (i.e., in the liver), does not inhibit the hepatic glucose production as well as it increases the glycolysis with subsequent conversion of carbohydrates to TG [49]. Moreover, IR also causes effects at the peripheral level: in muscle, it reduces glucose uptake; in adipose tissue, lipolysis increases with consequent high circulating levels of free fatty acids (FFA) used by the liver for TG synthesis [50,51]. Concerning DNL, BPA is able to increase de-novo fatty acid synthesis in Hep-G2 cells, by stimulating the activity of the enzymes involved in this metabolic pathway and, at the same time, up-regulating the expression of genes involved in lipid synthesis [12,13].

Both mechanisms, direct and indirect ones, induce the formation of lipotoxic lipid intermediates that contribute to the disruption of lipid homeostasis [13]. Some scientific evidence suggested the possible direct role of BPA in the induction of liver TG accumulation in animal models. In this regard, pregnant rats exposed to 50 μg/kg BW/day of BPA, orally administered, showed a reduction in the levels of the essential polyunsaturated fatty acid linoleic acid, evaluated by the analysis of TG contained in their liver [52]. Interestingly, long life non-obese BPA exposed mice developed modifications in lipid homeostasis, showing hepatic TG accumulation with distinct features from diet-induced obesity [53]. In particular, the exposure to environmental contaminants shares some mechanisms of induction of transcription of genes involved in hepatic lipid homeostasis with a high-fat high-sucrose diet acting, however, in a different but synergistic way with the high-calorie dietary regimen in stimulating the liver accumulation of TG [53]. Epigenetic regulation has become part of the pathophysiological process able to explain the relationship between exposure to BPA and hepatic TG accumulation. In HepG2 cells, the reduced expression of miR-192, and the increased activity of Sterol regulatory element-binding protein (SERBF1) and other genes involved in lipogenesis was correlated with the exposure to this environmental contaminant, finally leading to DNL induction. BPA would act in this context inducing an abnormal processing of pre-miR-192 by the class II ribonuclease enzyme DROSHA, which would have the role of blocking the DNL and the hepatic TG accumulation by acting on the 3’ untranslated region (UTR) of SREBF1 [54]. 

The role of BPA in the induction of steatosis, however, is not limited to the initiation of signal cascades involving hepatocytes. One of the peculiarities of this EDC should be to generate an anomalous activation of other cell types such as Kupffer cells, polarizing their differentiation into the pro-inflammatory M1 phenotype (M1KCs). This type of differentiation, induced by exposure to BPA should be able to regulate the expression of genes involved in lipid homeostasis, increasing both the hepatic lipid uptake and the DNL and contributing, moreover, to the elaboration of proinflammatory cytokines that represent the pathogenetic link through which BPA is able to induce the worsening of liver disease, trigger inflammation, fibrosis, and cirrhosis. In this scientific evidence the role of estrogenic receptors should seem of primary importance as demonstrated by the fact that this differentiation was blocked by estrogen antagonist ICI182780 use [53]. 

Furthermore, it has been recently demonstrated that the exposure of Hep-G2 cells at 0.025 and 0.05 μM of BPA for 72 h causes an increase in the intracellular quantity of TG evaluated with oil red O (ORO) staining microscopy and ORO colorimetric assay. This increase, although not extremely high, occurred in in vitro experimental conditions, where the role given by the absence of a hepatic cellular microenvironment composed of various other cell types that could intervene in the regulation of this process is not considered. This, in fact, might have possibly amplified the response related to the exposure to this environmental contaminant, worsening its accumulation and allowing the progression of hystological picture towards inflammation and fibrosis. Furthermore, the relatively short time of observation could be one of the explanations for which the researchers did not obtain a significant induction effect on hepatosteatosis [24]. 

### 3.2. Inflammation and Oxidative Stress: NAFLD-NASH Progression

In BPA-related NAFLD onset and progression, various pathogenetic mechanisms, apart from lipid hepatocellular accumulation, seem to be involved. Among them, the oxidative stress has a predominant role, strictly linked to the triggering of the inflammation cascade, that is a well-known factor able to initiate and sustain hepatocellular injury [55]. In this regard, interestingly, in human Hep-G2 hepatoma cells, BPA acts as a pro-steatotic and pro-inflammatory compound. In fact, in these cells, low-dose BPA exposition causes high production of ROS together with mitochondrial dysfunction and lipoperoxidation [55]. In particular, BPA-induced FFA accumulation in liver cells determined an overload of the metabolic pathway leading to their β-oxidation in the mitochondria, resulting in the formation of ROS and mitochondrial dysfunction due to the consumption of NADP [56]. In this regard, we demonstrated in a previous study that Hep-G2 cells exposed to BPA were characterized by an increase of oxidative stress and lipoperoxidation assessed by the evaluation of thiobarbituric acid reactive substances (TBARS) [24]. Interestingly, in view of a possible correlation with IR and diabetes, these latter effects were observed only in conditions of hyperglycemia (i.e., only at high concentrations of glucose in vitro). This finding could be explained considering the high amount of glucose as an energy source capable of supporting the energy demand to maintain cell proliferation and inflammation [24]. In this regard, it should be taken into account that patients with NAFLD have often metabolic comorbidities such as T2DM and obesity [57]. It should therefore be emphasized that the NAFLD patients suffering comorbidities probably represents a category of subjects at higher risk to meet all the harmful consequences deriving from exposure to BPA. In fact, patients with a greater quantity of visceral fat have a greater deposit site for BPA, due to its bioaccumulation in adipocytes [24,57]. Moreover, in the same study, NASH patients showed higher TBARS levels in comparison to NAFLD ones, suggesting the oxidative stress main role in NASH progression and the possible use of serum oxidative status as a predictive marker of response to an antioxidant treatment [24,58]. In fact, by the evaluation of the response to a therapy with silybin conjugated with phosphatidylcholine for 12 months, NASH patients that showed higher levels of TBARS at baseline demonstrated major improvement at the end of therapy in comparison to those with a lower level of TBARS at the beginning of the study. In this subset of the population acting on oxidative stress it was possible to reduce the endothelial dysfunction that represents the harmful link between NAFLD and cardiovascular disease [59]. ROS and lipoperoxidation products are able to increase the generation of several inflammatory cytokines playing a key role in cell death and inflammation [56]. This last one is not an irrelevant aspect: locally it could probably trigger the progression from NAFLD to NASH, but it may be related to systemic effects too. Cytokine release, as well as the fact that the BPA-induced FFA accumulation in liver cells determines directly a chronic inflammatory state, contributes to create a pro-inflammatory microenvironment [60,61].

The key passage from NAFLD to NASH could not only be the augmented oxidative stress and the lipoperoxidation, but also a reduction of cellular defense systems. In this regard in BPA exposed animals a down regulation of antioxidant enzymes activity (SOD, glutathione, CAT, reductase and peroxidase) and higher levels of oxidative stress markers (MDA and H_2_O_2_) have been found [13]. Normally, in a condition of oxidative stress, these enzymes are expected to be compensatory over expressed; instead, in this case their lower levels could suggest a possible enzyme system dysfunction or a down regulation gene expression BPA-induced. Another extremely promising area of research in this setting is the role of intestinal microbiota in the evolution of NAFLD. Recalling the concept of the pleiotropic effect of BPA in inducing and worsening NAFLD, scientific evidence has identified a link between exposure to BPA and homeostasis of intestinal microbial composition as well as intestinal permeability. Exposure to increasing doses of BPA has been correlated with a progressive increase of intestinal permeability, link to epithelial tight junction sealing, which would allow greater entry of Pathogens Associated Molecular Patterns (PAMPs) into portal blood and ultimately substantially increase the contribution to hepatic inflammation through inflammosome activation [62,63]. The effect of BPA, however, should not be limited exclusively to the regulation of intestinal permeability. Exposure to this environmental contaminant should also be able to alter the intestinal microbial composition above all by determining a reduction in the abundance of Short Chain Fatty Acids (SCFA)-producing bacterial species such as *Oscillospira* and *Ruminococcaceae*. Furthermore, this result would be associated with a clear reduction of SCFA in feces as well as an increase in Lipopolysaccharides (LPS) in portal and systemic blood. The in vitro addition of SCFA has been correlated with a substantial reduction of intestinal permeability which would suggest their key role in this setting [62]. Furthermore, the exposure to BPA assumed through food should have a direct effect in inducing inflammation of the intestinal wall through the stimulation of signal transduction pathways not yet fully identified, a condition that contributes decisively to the already complex BPA-related pathogenetic cascade, worsening the leaky-gut syndrome [64]. The worsening of liver disease, then, with the establishment of hepatic cirrhosis and portal hypertension could be itself capable of increasing the level of intestinal permeability, leading to the closure of a vicious circle through which the liver disease could progress in more advanced stages [65]. 

### 3.3. NASH/HCC Progression

BPA could probably also influence the HCC development. In fact, scientific evidence demonstrated that the exposure to BPA is able to induce DNA and chromosomal damage [66,67,68]. In vitro studies reported that BPA should be able to form DNA adducts in exposed cells and induce DNA strand breaks, as well as in vivo evidence described structural and numerical chromosomal aberrations after BPA exposure [69,70]. In this field, the increase in the expression of γH2A Histon Family Member X (γH2AX), used as a marker of DNA double strand breaks, assessed by flow cytometry, was detected in Hep-G2 cells exposed to 20 μg/mL BPA for 72 h [71]. The Hep-G2 mRNA expression of several genes involved in the xenobiotic metabolism (Cytochrome P450, family 1, subfamily A, polypeptide 1—CYP1A1, UDP-glucuronosyltransferase 1-1—UGT1A1, Glutathione S-transferase A1—GSTA1), oxidative stress (Glutamate–cysteine ligase catalytic subunit—GCLC, Glutathione peroxidase 1—GPX1, glutathione-disulfide reductase—GSR, superoxide dismutase 1A—SOD1A, CAT) and DNA damage response (tumore Protein 53—TP53, Mouse double minute 2 homolog—MDM2, cyclin-dependent kinase inhibitor 1—CDKN1A, GDD45A, Checkpoint Kinase 1—CHEK1, ERCC4) was also investigated by the same research group after 24 h of BPA exposure. The experiments highlighted the power of BPA and BPA-related analogues in the induction of these genes, demonstrating a possible role of these contaminants in oncological context [71]. Moreover, it has been also shown that BPA exposure can increase the cellular replication rate of HepG2 cells, but only in presence of high glucose levels [24]. Particular attention should be paid in this sense to that subgroup of patients affected by NAFLD with concomitant T2DM which, as already specified several times, represent the subset of the population at higher risk from BPA exposure. 

In a murine model, a dose-dependent correlation between BPA exposure and the incidence of HCC was demonstrated, with an odds ratio of 7.23 (95% CI: 3.23, 16.17) for the appearance of neoplastic and preneoplastic lesions in the liver of mice exposed to 50 mg BPA/kg diet, compared with unexposed controls [72]. If these findings are confirmed, polarizing also to the identification of signaling pathway involved in this response, BPA exposure could be recognized as an important risk factor for oncological diseases, in particular in T2DM patients. In this setting, once again, the triggering of inflammation and oxidative stress linked to exposure to BPA seems to play a central role. The addition of silybin in Hep-G2 cells cultures exposed to BPA should be able to block the effects of this contaminant, reducing the cellular proliferation and the oxidative stress. as confirmed by Western blot analyses of phosphorilized extracellular signal-regulated kinase (ERK) and Caspase 3, as well as spectrophotometry quantification of TBARS [12]. Silybin ameliorated the harmful effect of BPA by decreasing glucose uptake and lipid peroxidation. Moreover, silybin activated the synthesis of vitamin D3 metabolites and prevented the estrogens oxidation that, in turn, could be considered another important factor involved in NAFLD/obesity-associated HCC development [12]. 

## 4. Conclusions

The increased spreading of EDCs in daily common products, together with the increase of metabolic diseases, have directed scientists’ attention to the characterization and definition of the mechanisms of action of these compounds. 

The biological action possessed by this category of compounds seems to take place through various types of molecular interaction, a condition that makes it extremely difficult to study their key role in human pathology. On the other hand, the increasing daily exposure and growing evidence demonstrating their crucial role in the induction or worsening of metabolic pathology has led scientists from all over the world to pay particular attention to this group of substances.

In this regard, various in vitro and animal studies suggested a multifactorial involvement of BPA, not only in the onset of NAFLD, but also in its progression, due to the pleiotropic action of this EDC on the key factors involved in the pathophysiology of this disease. Therefore, more studies, in particular in clinical contexts, will be needed to identify a possible “risk management plan” related to BPA exposure, in order to give to the scientific community important information on the weight of the environmental factors on NAFLD pathogenesis and to give useful instruments to prevent metabolic diseases onset.

## Figures and Tables

**Figure 1 ijerph-16-03134-f001:**
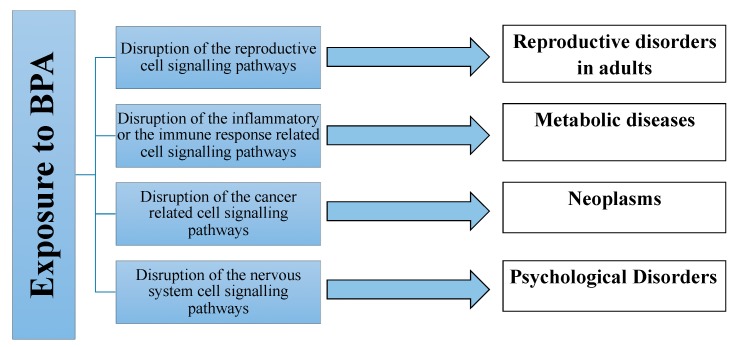
Main clinical bisphenol A (BPA) exposure-associated diseases.

**Figure 2 ijerph-16-03134-f002:**
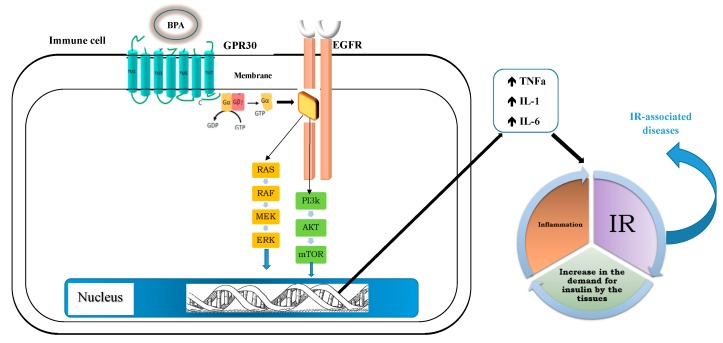
BPA-mediated G protein-coupled receptor 30 (GPR30) activation and relative effects.

**Figure 3 ijerph-16-03134-f003:**
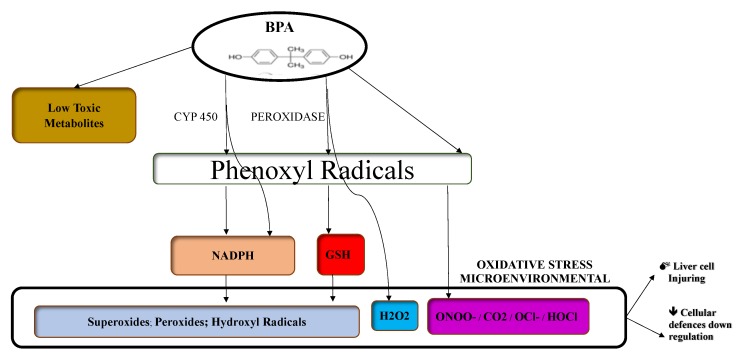
BPA-induced oxidative stress mechanisms.

**Table 1 ijerph-16-03134-t001:** List of main endocrine disruptors.

Endocrine Disruptors		
Arsenic	Insecticides	Fire retardants
Atrazine	Polychlorobiphenyl	Estradiol
Bisphenol A	Cadmium	Estrone
Lead	Parabens	Fungicides
Mercury	Pesticides	Perchlorate
Phytoestrogens	Bis (2-ethylhexyl)phthalate	Triclosan
Glycol ethers	Polycyclic aromatic hydrocarbons	Perfluorinated chemicals

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
