# Peer review of "Chemical Effect of Bisphenol A on Non-Alcoholic Fatty Liver Disease"

_ijerph, 2019, doi:10.3390/ijerph16173134_

Round 1

Reviewer 1 Report

Dallio M. and co authors extensively examined the possible role of Bisphenol A exposure on onset, progression and decompensation of NASH. The review is well written and represents an interesting reference for future studies. I have only one minor comment, Figure 1: I would change “Low levels of BPA” with “Exposure to BPA”

Author Response

Reviewer 1 comments: 

Dallio M. and co authors extensively examined the possible role of Bisphenol A exposure on onset, progression and decompensation of NASH. The review is well written and represents an interesting reference for future studies. I have only one minor comment, Figure 1: I would change “Low levels of BPA” with “Exposure to BPA” 

Response to reviewer 1: we thank the reviewer for her/his comment. As suggested we modified the sentence “low levels of BPA” with “Exposure to BPA”.

Reviewer 2 Report

MS Title: Chemical Effect of Bisphenol A on Non-Alcoholic 2 Fatty Liver Disease

This is an interesting manuscript on a very frequent chronic liver disease such as non-alcoholic fatty liver disease (NAFLD) and its relationship with endocrine disruptors (EDCs). I have some comments.

Page 1. Abstract. Line 2. I suggest to change the word hepatopathy by chronic liver disease

Page 1. Abstract. Line 3. I suggest to change the word complex. It has been written twice in the same line

Pages 1-2. Introduction section. I suggest to start this section with all information related with NAFLD and then…

Pages 1-2. Introduction section. I suggest to design a table with a list of the Endocrine disruptors (EDCs). Also for the readers it will important a figure to show how the EDCs interact with some physiological molecular signaling pathways.

Page 2. 1.1. The exposition and pharmacokinetics of BPA. First and second paragraphs. I suggest to give a worldwide overview of the exposition. But not only from the US.

Pages 1-2. Introduction section. In the end of this section it is important to include the outlines of this review

Page 6. Lines 232 and 235. In this regard… is redundant

Page 11. Conclusions I suggest to rewrite it. In its present form sounds redundant. The last five lines are ok.

The legend figures need a complete description. But not only he title

A list of abbreviations mandatory ns

Author Response

Reviewer 2 comments: 

MS Title: Chemical Effect of Bisphenol A on Non-Alcoholic 2 Fatty Liver Disease

This is an interesting manuscript on a very frequent chronic liver disease such as non-alcoholic fatty liver disease (NAFLD) and its relationship with endocrine disruptors (EDCs). I have some comments.

Page 1. Abstract. Line 2. I suggest to change the word hepatopathy by chronic liver disease

Page 1. Abstract. Line 3. I suggest to change the word complex. It has been written twice in the same line

Pages 1-2. Introduction section. I suggest to start this section with all information related with NAFLD and then…

Response to reviewer 2: we thank the reviewer for her/his comment. As suggested we completed all the above mentioned requests.

Pages 1-2. Introduction section. I suggest to design a table with a list of the Endocrine disruptors (EDCs). Also for the readers it will important a figure to show how the EDCs interact with some physiological molecular signaling pathways.

Response to reviewer 2: we thank the reviewer for her/his comment. As suggested we designed the table with a list of EDCs. Unfortunately we are not able to design a descriptive figure of action mechanisms of all the EDCs and this is the reason why we polarized our attention on the molecular action of BPA, as properly described in figure 2 and 3. A complete figure that shows the molecular mechanisms of all the EDCs could be extremely large and complex to understand, this is the reasons why we decided to not design this type of graphical support, also because, as suggested by the title of the manuscript, we manly talked about the role of BPA in NAFLD.

Page 2. 1.1. The exposition and pharmacokinetics of BPA. First and second paragraphs. I suggest to give a worldwide overview of the exposition. But not only from the US.

Response to reviewer 2: we thank the reviewer for her/his comment. We are completely agree with the reviewer. Unfortunately it is impossible to give this type of analysis because of lack of data published in scientific literature regarding this aspect. The only relative strong and recent data comes from US.

Pages 1-2. Introduction section. In the end of this section it is important to include the outlines of this review

Page 6. Lines 232 and 235. In this regard… is redundant

Response to reviewer 2: we thank the reviewer for her/his comment. As suggested we completed all the above mentioned requests.

Page 11. Conclusions I suggest to rewrite it. In its present form sounds redundant. The last five lines are ok.

Response to reviewer 2: we thank the reviewer for her/his comment. As suggested we rewrote the conclusion section.

The legend figures need a complete description. But not only he title

Response to reviewer 2: we thank the reviewer for her/his comment. We noticed that the caption of the figures were insert in the main text as error. We corrected the mistake.

A list of abbreviations mandatory ns

Response to reviewer 2: we thank the reviewer for her/his comment. We added the list of abbreviations at the end of manuscript.
